# Vegetation Change and Its Relationship with Climate Factors and Elevation on the Tibetan Plateau

**DOI:** 10.3390/ijerph16234709

**Published:** 2019-11-26

**Authors:** Yixin Zhang, Guoce Xu, Peng Li, Zhanbin Li, Yun Wang, Bin Wang, Lu Jia, Yuting Cheng, Jiaxin Zhang, Shaohao Zhuang, Yiting Chen

**Affiliations:** 1State Key Laboratory of Eco-hydraulics in Northwest Arid Region of China, Xi’an University of Technology, Xi’an 710048, China; leslie_cheung0707@163.com (Y.Z.); lipeng74@163.com (P.L.); zhanbinli@126.com (Z.L.); wenwu9264@163.com (B.W.); XiaMuXingYi94@163.com (L.J.); chengyutingstar@163.com (Y.C.); 18309294517@163.com (J.Z.); zsh437455842@163.com (S.Z.); yitingyoyo@gmail.com (Y.C.); 2Key Laboratory of National Forestry Administration on Ecological Hydrology and Disaster Prevention in Arid Regions, Xi’an University of Technology, Xi’an 710048, China; 3Digital Learning Resource Center, Hebei Radio and TV University, Shijiazhuang 050080, China; bingyu698@163.com

**Keywords:** climatic factors, Hurst exponent, vegetation fractional coverage

## Abstract

As the “roof of the world”, the Tibetan Plateau (TP) is a unique geographical unit on Earth. In recent years, vegetation has gradually become a key factor reflecting the ecosystem since it is sensitive to ecological changes especially in arid and semi-arid areas. Based on the normalized difference vegetation index (NDVI) dataset of TP from 2000 to 2015, this study analyzed the characteristics of vegetation variation and the correlation between vegetation change and climatic factors at different time scales, based on a Mann–Kendall trend analyses, the Hurst exponent, and the Pettitt change-point test. The results showed that the vegetation fractional coverage (VFC) generally increased in the past 16 years, with 60.3% of the TP experiencing an increase, of which significant (*p* < 0.05) increases accounted for 28.79% and were mainly distributed in the north of the TP. Temperature had the largest response with the VFC on the seasonal scale. During the growing season, the correlation between precipitation and sunshine duration with VFC was high (*p* < 0.05). The change-points of the VFC were mainly distributed in the north of the TP during 2007–2009. Slope and elevation had an impact on the VFC; the areas with large vegetation change are mainly distributed in slopes <20° and elevation of 3000–5000 m. For elevation above 3000–4000 m, the response of the VFC to precipitation and temperature was the strongest. This study provided important information for ecological environment protection and ecosystem degradation on the Tibetan Plateau.

## 1. Introduction

Climate change is receiving increasing attention, and governments and scientists are committed to researching and addressing this issue [1,2]. The Tibetan Plateau (TP), with an average elevation of over 4000 meters, which has a great heterogeneity and geographical variation of land cover, occupies five temperature zones and four water zones [3]. As the “Third Pole of the Earth”, It plays an important role in global climate change. According to current research, climate change has significantly affected many natural ecosystems [4,5,6], and it showed that the TP has one of the strongest climate warming signals in the world [7,8].

Vegetation plays an important role in the Earth’s material and energy exchange, which is the most sensitive part of ecosystems to climate change [9,10]. Vegetation can affect the surface radiation, soil quality, water cycle, and carbon cycle process, so it can indirectly regulate and stabilize the climate [11,12,13,14,15,16]. The vegetation distribution on the TP is complex and diverse. The horizontal belt includes forests, grasslands, meadows, etc. The vertical belt includes evergreen broad-leaved forests, coniferous forests, permanent alpine snow, and ice belts. It is the most complete vertical landscape of mountain landscapes in China [17]. In addition, the TP is also China’s largest ecologically fragile area. The monitoring results show that the problem of desertification in the plateau has become increasingly prominent [18]. With the development of remote sensing, high-resolution, long-term data can be used to study the core information of vegetation [19,20]. The normalized difference vegetation index (NDVI) can be applied to high spatial vegetation fractional coverage (VFC) and has become a key tool for analyzing VFC change dynamics history, monitoring current status, and predicting the future [21,22,23].

At present, most studies believe that average temperature and precipitation play an important role in the phenological changes of the TP [24,25]. The large spatial scale mainly based on observations of meteorological stations mainly include temperature, precipitation, sunshine duration, and relative wind speed [26]. In recent years, research on vegetation with climate on TP has gradually increased. The studies on the annual scale began earlier and relatively abundantly. Zhou et al. (2008) [27] reported that during 1998–2008, the correlation between temperature and vegetation change was high in the eastern and western parts of the TP, and in the central region, the correlation between temperature and vegetation was very strong. Peng et al. (2016) [14] used the Hurst index for 2000–2016 and pointed out that the vegetation on the TP had a growing trend. Xu et al. (2013) [28] reported that the simulation of vegetation indicated significant increases. Pang et al. (2015) [29] analyzed the relationship between precipitation and vegetation on TP from 1982 to 2012 and pointed out that the impact of precipitation on vegetation presents a complex relationship in partial areas. Zhuo et al. (2018) [30] reported that for 2000–2016, the area of increased vegetation on the TP was larger than the degraded area and pointed out that temperature has a significant effect on vegetation. Wang et al. (2015) [31] reported that sunshine duration in southeastern TP was possibly affected by large-scale ocean-atmosphere circulations and is as essential as temperature and precipitation for tree growth in TP. Sun et al. [32] studied the vegetation and climate on the seasonal scale from 1982 to 2013, and suggested that precipitation may have a hysteresis effect on vegetation. Most previous studies have discussed the relationships between vegetation and climate variables at the annual time scale and consider temperature to be the dominant factor for vegetation change in TP. However, there has been several studies on other time scales, such as the seasonal scale, dry–wet scale, and the growing season scale, especially in the spatial relationship of correlations. Since some studies are convinced that precipitation has different effects on the growth state of vegetation in different seasons [33,34], and have shown that climatic factors may change the growth season length of TP [35,36], these time scales should be considered. At the same time, relevant research shows that the variation of vegetation in the vertical direction is different, and the climate change in high-elevation areas has a greater impact on the vegetation ecosystem [37]. Attention should be paid to the differences between vegetation changes and climatic factors at different elevation. The discovery of these factors will provide guidance for vegetation management and restoration on the TP.

We aimed to explore the characteristics of vegetation changes for 2000–2015 on the TP; and to analyze the effects of climatic factors on the spatio-temporal growth of vegetation on multiple time scales, which is essential for the ecological stability of the region. Therefore, the study objectives were to: (1) Analyze the spatio-temporal changes characteristics of the VFC at the inter-annual scale; (2) analyze the relationships between vegetation and climate factors in different time scales; (3) discuss the causes of the VFC change-point in the research period; and (4) discuss the relationship between vegetation and climatic factors in the vertical direction.

## 2. Materials and Methods

### 2.1. Study Area

The Tibetan Plateau is mainly distributed in China, and is partially distributed in Bhutan, Nepal, India, and other countries; this study selected the parts of China as research objects (Figure 1) and the area is about 2,610,000 km², accounts for about 27.1% of China’s land area, which located in Central Asia and southwestern China (26°00′–39°47′ N, 73°19′–104°47′ E), and has an average elevation of over 4000 m. The basic features of TP are that solar radiation is strong, for 5000–8500 MJ/m^2^ per year. The average temperature reduced from 20 °C in the southeast to below −6 °C in the northwest, with a large temperature difference between day and night. Precipitation varies greatly among regions, with annual precipitation on the TP gradually declining from the southeast (>2000 mm) to the Qaidam Cold Lake Basin (<17 mm). In general, temperature and precipitation decrease from the southeast to the northwest, with climate in the southeast warm and humid and that in the northwest dry and cold. The dry and wet seasons are distinct. The typical soil of TP is meadow soil, mainly distributed in the east and southeast. The soil is thin and frozen for a long time in the Altay Mountains, the western Junggar Basin, and the Tianshan Mountains. The land types of TP are mainly cold and dry with low production capacity, the deserts and other hard-to-use land of TP are vast and represent >30% of the total area. The typical vegetation of the TP is alpine grassland, which supports the plateau animal husbandry. Under the combined effects of the natural environment and human factors, the ecological environment of the TP is extremely fragile [38].

#### 2.1.1. Study Area Division

The eco-geological division dataset of China was provided by the Data Center for Resources and Environmental Sciences, Chinese Academy of Sciences (RESDC) [39].

The basis of division is mainly temperature, moisture conditions, vegetation type, and soil type [40], and the TP was divided into 15 regions with totally different natural attributes (Figure 2 and Table 1).

#### 2.1.2. Division of Study Period

Four time scales were considered when analyzing the correlation between vegetation and climatic factors, which includes annual scale, seasonal scale, growing season, and dry–wet season scale. Seasonal is consistent with meteorological standards, which are spring (March–May), summer (June-August), autumn (September–November), and winter (December–February). Dry–wet season scale division was based on monthly average precipitation, which includes wet season (May–September) and dry (October–April) season. The growing season is March–November.

### 2.2. Data Preparation

The two types of datasets used were NDVI and meteorological data (precipitation, temperature, and sunshine duration). The data coordinate system was set to D-WGS-1984, and the pixel precision was unified to 0.02° × 0.02°.

The NDVI data were obtained from a long-term moderate-resolution imaging spectroradiometer (MODIS) vegetation index dataset, with spatial resolution of 500 m and 10–day interval, for the period February 2000 to December 2015. The annual NDVI data used a maximum value composite method. The data were provided by Geospatial Data Cloud site, Computer Network Information Center, Chinese Academy of Sciences [41].

Climate factors included in this study were precipitation, temperature, and sunshine duration. We selected 147 of the 824 basic meteorological stations operating since 1951 in China. Consistent with the period of the NDVI data, the monthly climate dataset (V3.0) was downloaded from 2000 to 2015. Thirty-eight meteorological stations in Tibet, 33 in Qinghai Province, 29 in Xinjiang Province, 15 in Gansu Province, 8 in Yunnan Province, and 22 in Sichuan Province. The datasets were provided by the China Meteorological Data Sharing Service Network [42].

### 2.3. Method

#### 2.3.1. Linear Regression Analysis, Mann–Kendall Trend Test, Change-Point Test, and Correlation Analysis

VFC is an important vegetation indicator that can be calculated by Equation (1):(1)$$VFC=\left(NDVI-NDVI_{min}\right)/\left(NDVI_{max}-NDVI_{min}\right)$$
where ($NDVI_{max}$) and ($NDVI_{min}$) represent the maximum and minimum of NDVI in the region, respectively.

The trend of vegetation coverage was calculated by Equation (2), which is widely used [28,29]:(2)$$P=\frac{n\times \sum_{i=1}^{n}\left(i\times VFC_{i}\right)-(\sum_{i=1}^{n}i\sum_{i=1}^{n}VFC)}{n\times \sum_{i=1}^{n}i^{2}-{\left(\sum_{i=1}^{n}i\right)}^{2}}$$
where P is the rate of vegetation coverage; n is the time series length; i  is the number of year, with a range of 1–16; $VFC_{i}$ is the vegetation coverage of the ith year. The change rate of vegetation coverage (E) was calculated using Equation (3):(3)E=P(n−1).

The Mann–Kendall trend test method is widely used to analyze time series trend changes of climatic factors [43]. Define statistic S and symbolic functions (sign) as Equation (4) and Equation (5):(4)$$S=\sum_{i=1}^{n-1}\sum_{j=i+1}^{n}sign\left(X_{i}-X_{j}\right)$$
(5)$$\mathrm{sign}\left(X_{i}-X_{j}\right)=\{\;\begin{array}{c} +1,\;if\;\left(X_{j}-X_{i}\right)>0\\ \;0,\;if\;\left(X_{j}-X_{i}\right)=0\\ -1,\;if\;\left(X_{j}-X_{i}\right)<0 \end{array}$$
(6)$$Z=\{\begin{array}{c} \frac{S-1}{\sqrt{n\left(n-1\right)\left(2n+5\right)/18}},if\;S>0\\ \;0,if\;S>0\;\\ \frac{S+1}{\sqrt{n\left(n-1\right)\left(2n+5\right)/18}},\;if\;S<0 \end{array}.$$

Statistic Z was calculated using Equation (6); when Z > 0, it shows an increasing trend, and when Z < 0, it is a decreasing trend. When the absolute value of Z is greater than 1.28, 1.64, and 2.32, it means that the confidence is 90%, 95%, and 99%, respectively.

The Hurst exponent is widely used to predict the future trend of time series [44].

A time series is defined as t = 1, 2, 3, …, *n*, and the mean sequence of the series is defined using Equation (7)
(7)$${\bar{f}}_{\left(\tau \right)}=\frac{1}{\tau }\sum_{t=1}^{\tau }f_{\left(\tau \right)\;\;}\tau =1,2\ldots .,n.$$

The cumulative deviation is calculated using Equation (8):(8)$$X_{\left(t,\tau \right)}=\sum_{t=1}^{t}(f_{\left(t\right)}-{\bar{f}}_{\left(t\right)}),\;1\leq t\leq \tau .$$

The range sequence is calculated using Equation (9):(9)$$R_{\left(\tau \right)}=\underset{1\leq t\leq \tau }{\max}X_{\left(t,\tau \right)}-\underset{\;1\leq t\leq \tau }{\min}X_{\left(t,\tau \right)},\tau =1,2\ldots ,n.$$

A standard deviation sequence is calculated using Equation (10):(10)$$S_{\left(\tau \right)}={\left[\frac{1}{\tau }\sum_{t=1}^{t}{\left(f_{\left(t\right)}-{\bar{f}}_{\left(t\right)}\right)}^{2}\right]}^{1/2},\;\tau =1,2\ldots .,n.$$

Finally, the Hurst exponent is calculated using Equation (11):(11)$$\frac{R_{\left(\tau \right)}}{S_{\left(\tau \right)}}={\left(c\tau \right)}^{H}.$$

When 0 < H < 0.5, the sequence under investigation is an anti-persistent sequence, and H = 0.5 indicates that the sequence under investigation is a random sequence; 0.5 < H < 1.0 indicates that the sequence under study is a persistent sequence. The closer the H value is to 1, the stronger the persistence is.

The Pettitt change-point test, which is widely used to identify and extract the corresponding maximum value in a time series in climate change, is calculated using Equation (12):(12)$$S_{k}=2\sum_{i=1}^{k}r_{i}-k(n+1);\;k=1,\;\ldots ,n$$
(13)$$S_{E}=\underset{1\leq k\leq n}{\max}\left|S_{k}\right|.$$

For VFC sequences $x_{1}$, $x_{2}$,... $x_{n}$ with a sample size of n, the corresponding order is listed as $r_{1},\;r_{2}$,... $r_{n}$; then, the statistics are constructed:$\;S_{k}\;$ is the test result. E is the year in which the change-point occurred.

#### 2.3.2. Calculation Environment

In this study, all calculations were performed on the pixel scales. For the sake of description, we refer to pixels images as grid. The Mann–Kendall trend test, the Pearson correlation test, Hurst exponent, and Pettitt test were performed in the R environment using the ‘trend’ (version 1.1.1), ‘stats’ (version 3.4.3), and ‘Random Fields’ (version 3.1.5) packages (RStudio Inc., Boston, MA, USA).

## 3. Results

### 3.1. Characteristics of Vegetation Coverage Statistics in Eco-Geological Zones

The VFC change of each eco-geographical zone is shown in Table 2. The VFC generally decreased from southeast to northwest. For 2000–2015, the mean VFC of the TP was 0.47, indicating that the VFC was still at a low level on multi-year. From the E exponent (Figure 3), the VFC growth of the total area accounted for is 60.3%, and 38.7% decreased. The significant increased area (*p* < 0.05) accounted for 28.7% of the total area from the M-K trend test (Table 3). The mean of the VFC was highest in the Central Asia Subtropical zone (VA5 and VA6), with 0.83, followed by the middle temperate (IID3 and IID5) and warm temperate (IIID1) zones, with a mean VFC of 0.55 and 0.40, respectively. The areas with relatively low VFC were in the plateau temperate (HIIAB1, HIIC1, HIIC2, HIID1, HIID2, and HIID3) and the plateau sub-frigid (HIB1, HIC1, HIC2, and HID1) zones, with means of 0.39 and 0.37, respectively.

### 3.2. Characteristics of Dynamic Change and Consistency of Trends in Vegetation Coverage

The E exponent showed low annual variation in vegetation (Figure 3), concentrated in the interval [–25%, 25%]. The positive growth area of VFC (2,000,000 km^2^) was much larger than the negative growth area (600,000 km^2^). The VFC with a small range of increase was mainly distributed in HID1, HIID1, and the high-elevation areas of HIID2 in the north of the TP. The VFC with a small range of decrease was mainly distributed in HIC2, HIIC2, HIB1, and H2A/B1, which are central and southern areas of the TP.

The M-K trend test effectively showed the trend of the VFC (Figure 4). During 2000–2015, 28.7% of the TP had a significant (*p* < 0.05) growth trend, only 0.47% of the TP had a significant decreasing trend (*p* < 0.05), 1.79% of the area remained relatively stable, and the remaining 68.2% of the TP had no significant change (*p* > 0.05). Table 3 showed that the area of the VFC with severe improvement (*p* < 0.001) accounted for 4.18%, mainly in the HID1, HIID1, and the high-elevation areas of HIID2 in the north of the TP. The area of moderate improvement (*p* < 0.01) was 10.71%, mainly in HIC2, HIC1, HIB1, VA5, and VA6, which is the central and southeast of the TP. The area of slight improvement (*p* < 0.05) accounted for 13.89% and the slightly degraded area accounted for 0.07%, which was mainly in HIIC2, HIB1, and H2A/B1. The moderate and severely decreased areas of the VFC were relatively small, and mainly in the Yili and Qaidam Basins.

The Hurst exponent (Figure 5) with the values >0.5 accounted for 96.63% of the entire TP and mainly distributed in the north, which indicated that the future development trend will be consistent with the current; the values >0.8 accounted for 30.63% of the area, which indicated a strong consistency. The values <0.5 indicated an inconsistency and only accounted for 4.3% of the entire region. Combined with the M-K test (Figure 4), it showed that the stronger the growth of VFC in the past, the stronger the trend of sustained growth in the future; the highly degraded regions will show a trend inconsistent with the past, that is, areas with historical severe vegetation degradation may recover in the future.

### 3.3. Seasonal Correlations between VFC and Climatic Factors

The trend of climatic factors on TP and surrounding areas are significantly different seasonally, especially for precipitation and temperature, which are closely related to vegetation change and can be mediated by the growth and distribution of vegetation, to a certain extent [45,46]. In this study, the correlation coefficients between the seasonal mean VFC and climatic factors (precipitation, temperature, and sunshine duration) were calculated to determine the seasonal response of climatic factors to the VFC. The seasonal mean VFC showed different spatio-temporal relationships corresponding to the changes of seasonal precipitation, temperature, and sunshine duration (Figure 6). The numerical statistics of the significant (*p* < 0.05) correlation area in each season, and the distribution of the significant response area in the eco-geographical partition, are shown in Table 4. In general, the correlation between precipitation and VFC is the strongest in summer, with positive correlation as the main, and weakest in autumn, with a negative correlation. Temperature and vegetation responded strongly positive in spring and autumn, and the weakest negative correlation in winter. Sunshine duration and vegetation responded strongly in summer and autumn.

### 3.4. Correlations between the VFC and Climatic Factors in Wet and Dry Seasons

Dividing dry and wet periods according to multi-year of monthly average precipitation, the average monthly precipitation is greater than 40 mm, which is defined as the wet season, and the rest is dry season. The precipitation in dry season is only 1/3 to 1/2 of the wet season, which leads to seasonal drought and thus affect vegetation growth, and the response of VFC to climatic factors can be identified under different water conditions [47]. In order to explore the relationship between the VFC and climatic factors (precipitation/temperature/sunshine duration), the Pearson correlation coefficients between average VFC and climatic factors were calculated for each grid-point in the dry and the wet season (Figure 7). The darker the color in the graph, the greater the density of the grid-points.

Precipitation was concentrated in the 0–5 mm interval in the dry season, but was concentrated in 0–80 mm in the wet season (Figure 7a,b). The micro-precipitation in the dry season was mainly negatively correlated with the VFC. In the interval of 0–20 mm in the wet season, with increased precipitation, the response of precipitation and the VFC rapidly changed from high negative to high positive. In the interval of 20–70 mm, precipitation and the VFC were always highly positive; for the precipitation >70 mm, the response was decreased.

Temperature in the dry season was concentrated in the interval of [–10, 10] °C, and the correlation between temperature and VFC did not show a particularly clear relationship, while in the wet season, temperature was mainly concentrated within [5, 15] °C (Figure 7c,d) and temperature always showed a high positive correlation with VFC in this interval.

Sunshine duration in the dry season was concentrated within 160–270 h, and at around 250 h, most of the grid-points of the VFC and sunshine duration were positively correlated (Figure 7e,f). The sunshine duration in the wet season was mainly concentrated in the interval of [150, 300] h, and near to 250 h, the VFC and sunshine duration showed a dense negative correlation region.

### 3.5. Response of Climate Factors to the VFC in the Growing Season

In the context of climate change, studies have pointed out that the growing season of the TP has advanced or delayed [48,49]. Although the conclusions are not completely consistent, the vegetation in the growing season is affected by the climate, and the high coverage area does move [50]. The response of the climatic factors to the VFC in the growing season has a certain reference role in the restoration and protection of vegetation in the TP. The correlations of precipitation, temperature, and sunshine duration with the VFC in the growing seasons for 2000–2015 were graded and superimposed according to significance (*p* < 0.01). Only significantly affected by one climate factor was termed a “single-factor” grid, affected by any two climate factors is a “double-factor” grid, and significantly affected by all three climate factors is a “triple-factor” grid (Figure 8). Areas with significant (*p* < 0.01) single-factor for precipitation were concentrated in the middle (HIC2) and northeastern (HIIC1) parts of the TP. Areas with a significant (*p* < 0.01) single-factor for sunshine duration were mainly in the north (HIID1). Areas with a single-factor grid for temperature was sporadically distributed in HIB1 and HIIA/B1, the southeast of the TP. Double-factor grids were mainly distributed at the junction of HIID1, HIIC1 in the northeast, the junction of HIC2, HIIC2, HIB1, and H2A/B1 in the central, and VA5 and VA6 in the southeast. Triple-factor grids were concentrated at the junction of HIID1, HIIC1, and HIC1.

## 4. Discussion

### 4.1. Vegetation Variation and the Distribution of Change-Points

During 2000–2015, the VFC increased from 0.39 to 0.43 throughout the whole region, which is inseparable from the effect of climate change on the TP [27,28]. The VFC as a whole seemed to be in a slow recovery process. In fact, >80% of the area’s VFC was still at a low level [51,52]. The integral VFC improvement is mainly dependent on local growth, instead of the entire region. Restoration of VFC was significantly better in the north, for example, the Kunlun high mountain plateau desert area (HID1), the desert area in the north wing of the Kunlun Mountains (HIID2), Qingdong Qilian high mountain basin (HIIC1), Ali mountain desert area (HIID3), and high-shrubland steppe (HIIC2) showed significant improvement during the study period (Figure 4). To further explore the inter-annual change-point of the VFC, a Pettitt change-point test was performed on the VFC sequence (Figure 9). The area with significant (*p* < 0.05) change-point during 2000–2015 accounted for 16.9% and was mainly distributed in the northern. Significant (*p* < 0.05) change-points were concentrated during 2007–2009. In 2007 and 2008, the significant change-point accounted for 5.3% and 5.9%, respectively, and in 2009, the change-point occurred most strongly, accounting for 6.5%. After 2009, the area of significant change-point dropped sharply. After 2010, there were almost no change-points in VFC of the entire region. In addition to climatic factors. the change-point of VFC in the northern region is inextricably linked to human ecological construction, especially in the northeast of the TP, which mainly includes Qinghai Province, Gansu Province, and part of Xinjiang and northern Tibet. These areas have implemented a variety of ecological engineering measures during the study period, such as the Three-North Shelterbelt Project, the Natural Forest Protection Project, and the Returning Farmland to Forests/Grassland Project [53,54]. Due to the implementation of ecological engineering measures in the northeast, the VFC center of gravity in the TP slowly migrated to the north (Figure 10). It is noteworthy that the Three-North Shelterbelt Project has a geographical boundary of 34–35° N in the southwestern China and the main objective of the ecological project in these areas was desertification control, although the ecological projects are not covered the entire region, and the governance effect was not as obvious as the other three natural regions in China, which are the eastern monsoon and the northwest arid and semi-arid region. However, compared with the vegetation under the natural conditions or single measure in the south of TP, the change was still obvious. The superposition of the ecological project matches the occurrence of the inter-annual change-point, the first phase of the Returning Farmland to Forests/Grassland Project in the north of TP began in 2000 and completed in 2005 [55], while the effects of ecological project seemed to be slow and had a time lag after the start of the first phase, and the center of vegetation gravity shifted significantly about two years later. The second phase of the ecological project was in 2006–2010. In addition, China has strengthened protection of the TP’s ecological core functional areas and prohibited development for multiple regions since 2008. Thus, during 2007–2009, a variety of ecological engineering and management measures were superimposed in this region, and the VFC change-points in the project range reached the maximum. After the end of the second phase in 2010, the change-point in the north began to weaken and disappear. From this point of view, the vegetation improvement in the north was largely related to human activities. However, it is worth noting that there is a time lag between the beginning and the greatest effect of ecological project; when the ecological project stopped, a rebound would soon appear, especially in areas with a poor vegetation base. Therefore, ecological projects in this area will not be once and for all and long-term and multi-vegetation restoration measures are effective and necessary [56]. Although approximately 150 nature reserves have been established in the TP, in comparison, there are few ecological projects implemented in the south, and the vegetation in southern areas is mostly controlled by a single nature reserve management, while most nature reserves are mismanaged or the management systems do not match, there is a lack of inspection and monitoring mechanisms [57], and the effect of vegetation improvement is not as obvious as it in the north.

### 4.2. Correlation between Vegetation and Climatic Factors and Their Vertical Differentiation

The analysis of past temperature and precipitation trends in the TP shows a trend of warm and humid, and some mathematical models are used for future climate prediction. Gao et al. (2016) [6] used the LPJ model for climate prediction of the TP and pointed that the future temperature will increase by 1.3–4.2 °C, and annual precipitation will increase by 2–5%. Furthermore, in the emergency context of global warming, research on climatic factors and vegetation change diversified, especially at the annual scale. Although the lengths of time series adopted by scholars were not completely consistent, they generally show that there was a strong positive relationship between NDVI and precipitation, especially in the northeastern plateau, and suggested that precipitation was a favorable factor for the grassland, while the negative correlations between NDVI and temperature, especially in the southern plateau [58,59]. For the seasonal scale, we found that when the time scale is shortened, the response intensity of climate factors to VFC in the whole region was not balanced in different seasons. Precipitation is more severe in summer, increased precipitation during the summer contributes to the growth of VFC, and the positive correlation areas are mainly distributed in the arid alpine desert area in the north, while the relevance depends mainly on whether the period of precipitation and heat is consistent [29]. The precipitation in spring inhibits the growth of vegetation in some areas, which is mainly distributed in colder and arid regions and may be affected by snow cover [60]; the VFC in the low-temperature area such as the Kunlun Mountains plateau is mainly controlled by temperature and related to the delay effect of precipitation on vegetation growth [48]. When the temperature of spring has not recovered significantly, the precipitation is not enough to increase the vegetation. In addition to winter, the response area of temperature to VFC is relatively large, and almost positive. Interestingly, the positive response area in spring and autumn is generally larger than it in summer, which is consistent with the sun (2016) [32], the precipitation and temperature have high average in summer, and the regulation of precipitation is very significant in summer (Figure 6a). In spring and autumn, the change of precipitation in most areas is very rapid, and during this period, temperature becomes the dominant climatic factor controlling vegetation growth [47,61]. The sunshine duration with VFC showed the strongest response in summer and winter. The northern warm temperate desert area (HIID1 and HIIC1) showed negatively correlated with VFC in summer, which means that the vegetation was not promoted with the increase of sunshine duration. Meanwhile, in the humid and sub-humid areas of the southwest, the increase in sunshine duration promoted vegetation growth. The opposite relationship between winter and summer may be related to the water stress, vegetation types, and potential evapotranspiration of each eco-geographic region [62]. The increase of temperature and sunshine duration in summer has certain inhibition on the vegetation in the arid desert area [63]. On the dry and wet season scale, the climatic factors are more sensitive to the change of VFC in the wet season. During the wet season, the increase of precipitation obviously promotes the vegetation with the range of 0–20 mm. In the range of 20–60 mm, precipitation still promotes vegetation growth, while 60 mm seems to be a turning point. When it is larger than this range, precipitation will inhibit vegetation growth. While the temperature is highly positive correlated with VFC in the wet season, when water stress is not considered, temperature is the main factor controlling VFC change in the TP. The sunshine duration in Figure 7 does not show a clear change from the VFC, probably due to altitude effects. In the dry season, under the condition of insufficient supply of precipitation, the change of VFC may be controlled by multiple climatic factors, so the dispersion in the figure is stronger. In the growing season, the semi-arid area (HIC2) only receives the influence of precipitation changes, which may be due to the influence of vegetation types. When the temperature rises, the grassland is more sensitive to the appropriate range of precipitation [64]. Humid and sub-humid areas eliminate water stress and temperature is critical to VFC [65]. In the northern Qaidam Basin (HIID1), due to the elevation, the monthly sunshine duration is lower than other areas, so that the sunshine duration control the VFC changes in the growing season. The effect of elevation on vegetation and climate factors does not only exist in this area.

Simulation and field experiments show that elevation affect the VFC of alpine vegetation, and with the change of elevation, the relationship between the VFC and temperature has different characteristics [66,67,68]. In the higher elevation areas in the northwest, the degree of human disturbance was relatively low, and climate was the main factors controlling of vegetation growth. The VFC at high elevation is relatively low compared to the middle and low elevation areas. However, from the perspective of the significance of vegetation changes, in high-elevation mountain areas such as the Kunlun high mountain plateau desert area (HID1) and the desert area in the north wing of the Kunlun Mountains (HIID2), the vegetation had a significant improvement (Figure 4). In low-elevation areas, human factors had a strong effect [29] and climate impacts became secondary, such as for the Yili Basin (IID5) and the Qaidam Basin (HIID1), which experienced significant degradation during the study period [69]. Tao (2013) [70] pointed out that the fact that the varying temperature change trends as regulated by elevation implies that temperate grasslands have experienced a more rapid temperature change than alpine grasslands during the past decade. Thus, the impact of elevation on vegetation and climate is worth to discuss. In this study, we mainly use elevation as the geographical element. Then, the correlation coefficient between elevation and VFC for 2000–2015 is shown in Figure 11. The VFC was negatively correlated with elevation over the years. VFC changes fluctuate at different altitudes, the large change (E > 50%) was mainly in the areas of elevation <1500 m and middle–high elevation around 5000 m (Figure 12). The change range of VFC in the area >6000 m was basically stable. At the same time, we found that for slope of 0–20°, the VFC had a large fluctuation range, and the VFC mainly increased. For slope >40°, the variation of the VFC gradually decreased with increased slope and was basically stable until 80° (Figure 13).

As can be seen from Figure 11, the degree of negative correlations gradually decreased during the study period, which indicated that as the elevation increases, the degree of VFC decrease was aggravated. For the entire region, the change with correlation coefficient in the vertical is noteworthy. To explore the difference in the value of correlation at different elevation levels, and segmentation analysis for every 1000 m of total elevation showed that the VFC was positively correlated with elevation for areas <2000 m (Figure 14). It is noteworthy that in low-middle elevation area (2000–3000 m), the temperature decreases as elevation increases, and the VFC may be affected by changes in temperature over the vertical gradient; the VFC is sensitive to temperature at this elevation segmentation and showed a trend of reduced as elevation increases, and this may also relate to human activities. In the middle elevation (3000–4000 m), the VFC was positive correlated with elevation; with increased elevation, the VFC tended to increase, and this positive correlation remained stable for years. Although the VFC was negative correlated with elevation >4000 m, it is worth noting that the region with the largest negative correlation coefficient is not the highest elevation level, and this may reflect the response of alpine vegetation to the environment under climate change, which is consistent with Tao et al. (2013) [70].

Climate has different effects on vegetation at different elevations level, and we analyzed the correlation between climatic factors (precipitation and temperature) in different elevation stratifications [71]. During 2000–2015, precipitation with the VFC were generally positively correlated in each stratification, and we noticed that in areas with elevation <5000 m, the correlation coefficient between precipitation and vegetation is large. When the elevation >5000 m, the positive correlation is significantly reduced (Figure 15). Temperatures at elevation <4000 m were positively correlated with VFC (Figure 16), especially at 2000–3000 m, where the positive correlations were the largest; at 4000–5000 m, correlations between temperature and the VFC changed from positive to negative, and the negative correlations were largest at 5000–6000 m. Combined with Figure 15 and Figure 16, in the low-elevation areas (<2000 m), human activities and other disturbances are enormous, and VFC was not sensitive to precipitation changes. As elevation increased, human disturbance declined, and the influence of precipitation on the VFC gradually became more sensitive. At relatively high elevations (>5000 m), solar radiation and temperature dominated and the relationship between precipitation and vegetation was weakened. Precipitation and temperature at elevations of about 3000 m had the greatest positive effect on the VFC. In fact, the area at about 3000 m accounted for 27% of the TP, and the average elevation of cities with large population density in the TP are basically in this elevation stratification, which including Lhasa (3650 m), Xining (3137 m), etc., and the average elevation is in this area. At this elevation, the response to vegetation restoration and climatic factors is sensitive and can be prioritized.

### 4.3. Limitations of This Study

The climatic factors used in this study were derived from meteorological stations. At present, because the meteorological observation sites are less distributed in the southwest of TP, the conclusions of this study have limited application in areas other than the southwest. The climatic factors we concerned were limited to precipitation, temperature, and sunshine duration. The relationship between other meteorological factors and vegetation changes were not explored in depth. In the future, the mutual feedback of various climate factors and VFC changes need to be considered, and the delayed effects of climatic factors on vegetation should be analyzed according to the climatic zone.

## 5. Conclusions

The overall characteristics of the VFC were “high in southeast and low in northwest” and generally at a low and medium level. The largest increase in the VFC was mainly in the northern Kunlun Mountains, and the most serious degradation was concentrated in the Qaidam and Yili Basins. The Hurst exponent indicated that future VFC trends are consistent with current. On the seasonal scale, precipitation, temperature, and sunshine duration were strongest in summer, autumn, and winter, respectively. During the growing season, precipitation and sunshine duration had a relatively large vegetation response area (*p* < 0.05). Temperature was more representative of the seasonal response, and precipitation and sunshine duration were more representative of the growing season VFC response. Different spatial distributions of significant regions at different time scales suggested that the relationship between VFC needs to adopt a multi-scale perspective. The change-point of VFC mainly distributed in the north of the TP and concentrated in 2007–2009. The negative correlation between VFC and elevation correlation coefficient was getting stronger. When the elevation is around 3000–4000 m, the VFC had the greatest correlation coefficient with precipitation and temperature, and this elevation range should be subject to further study.

## Figures and Tables

**Figure 1 ijerph-16-04709-f001:**
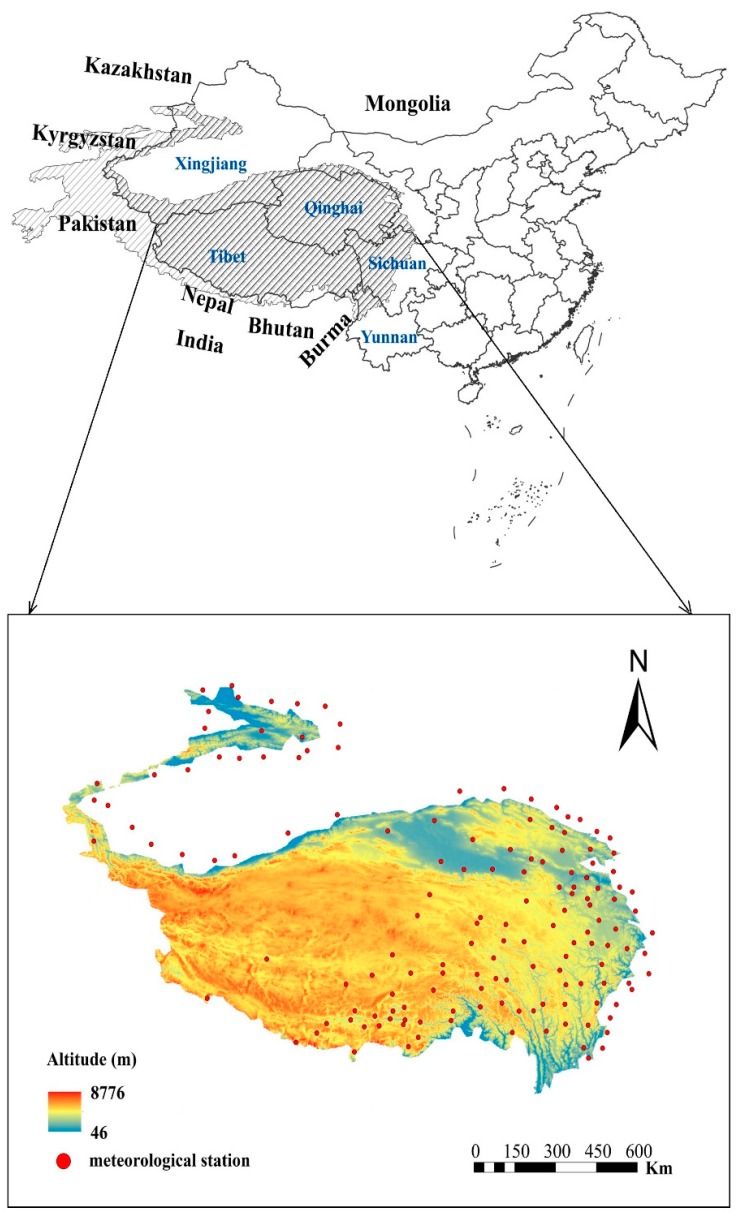
Location of study area.

**Figure 2 ijerph-16-04709-f002:**
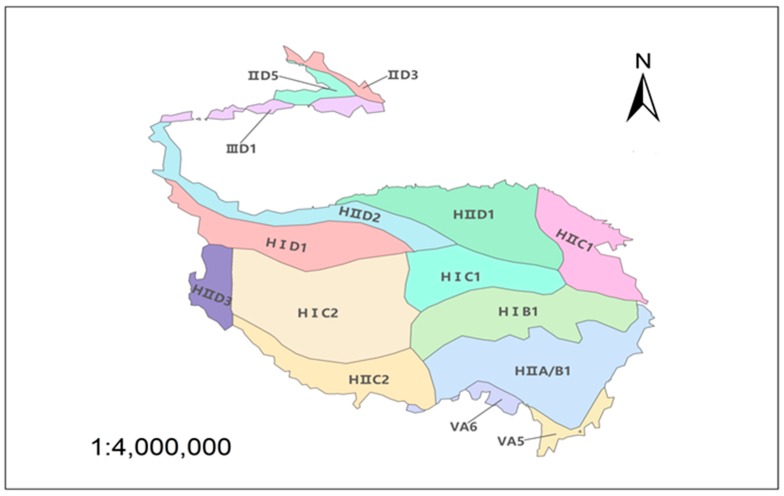
Eco-geological division of the Tibetan Plateau.

**Figure 3 ijerph-16-04709-f003:**
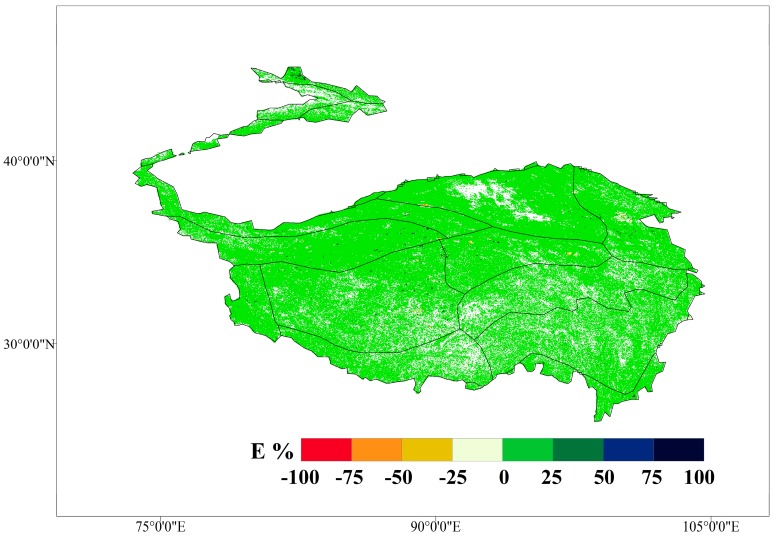
E exponent value of the Tibetan Plateau for 2000–2015.

**Figure 4 ijerph-16-04709-f004:**
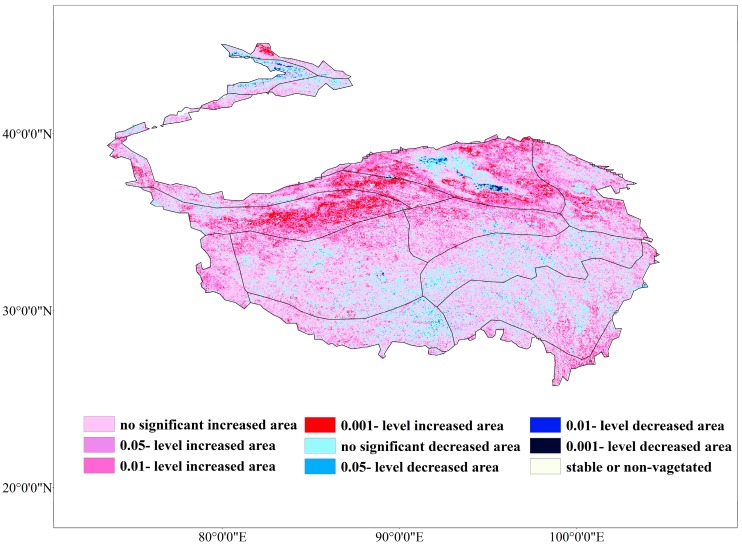
Different types of change trends in normalized difference vegetation index (NDVI) from 2000–2015.

**Figure 5 ijerph-16-04709-f005:**
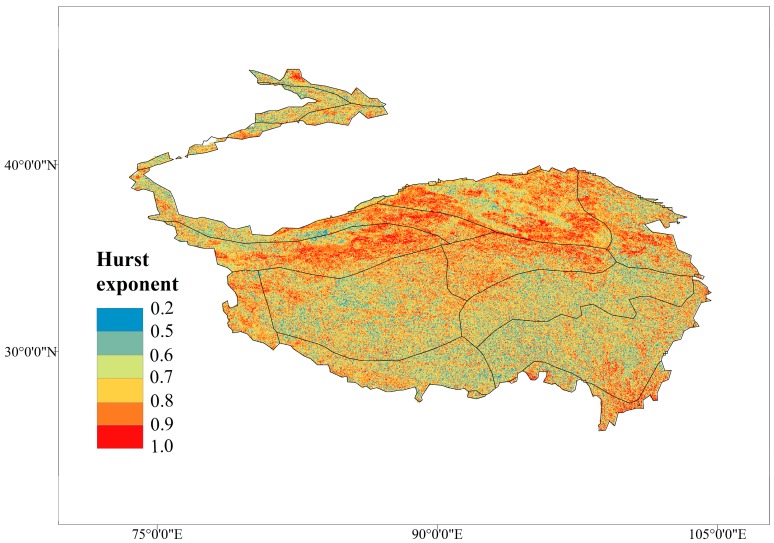
Hurst exponent of the Tibetan Plateau for 2000–2015.

**Figure 6 ijerph-16-04709-f006:**
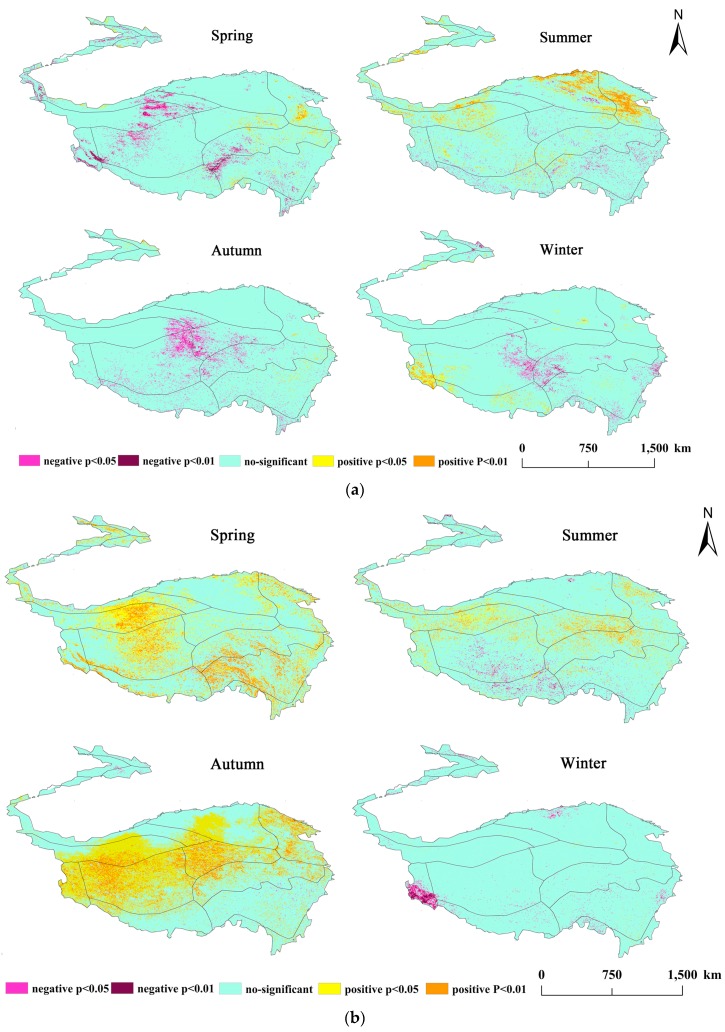
(**a**) Spatial distribution of correlation coefficients between the mean VFC and seasonal precipitation. (**b**) Spatial distribution of correlation coefficients between the mean VFC and seasonal temperature. (**c**) Spatial distribution of correlation coefficients between the mean VFC and seasonal sunshine duration.

**Figure 7 ijerph-16-04709-f007:**
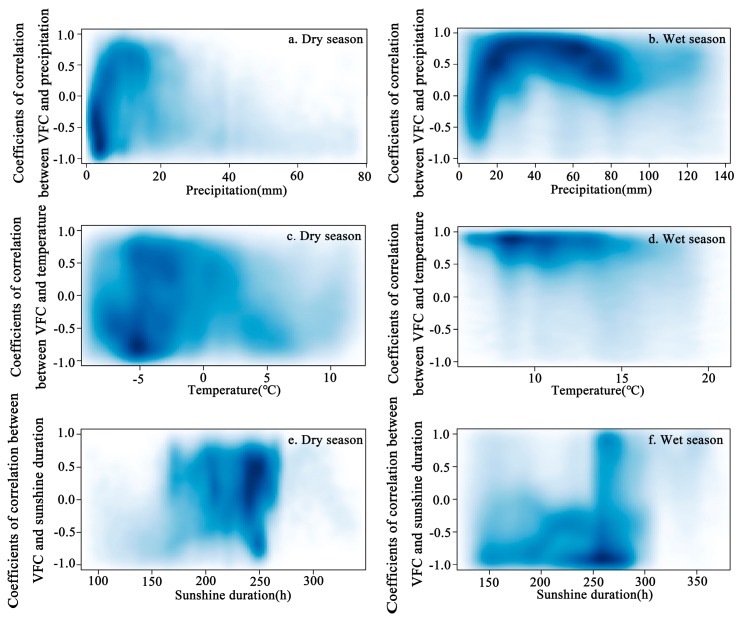
The correlation coefficients between mean the VFC and climatic factors in dry and wet seasons. (**a**) Correlations between the VFC and precipitation in dry season. (**b**) Correlations between the VFC and precipitation in wet season. (**c**) Correlations between the VFC and temperature in dry season. (**d**) Correlations between the VFC and temperature in wet season. (**e**) Correlations between the VFC and sunshine duration in dry season. (**f**) Correlations between the VFC and sunshine duration in wet season.

**Figure 8 ijerph-16-04709-f008:**
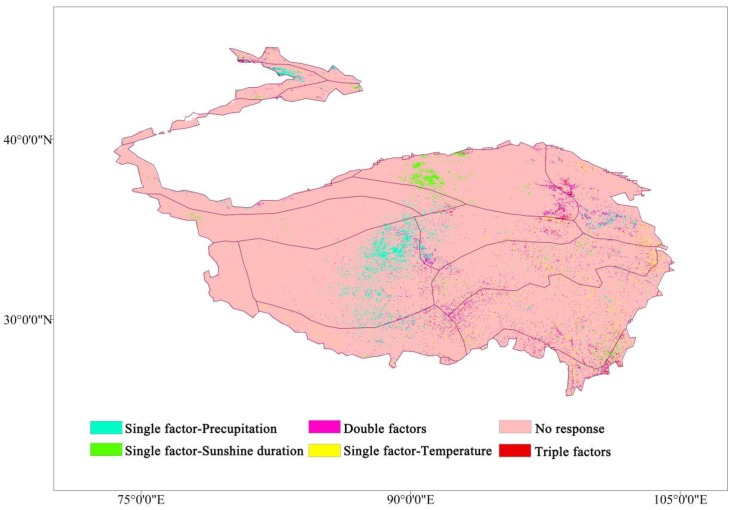
Response distribution of climatic factors to the VFC in the growing season.

**Figure 9 ijerph-16-04709-f009:**
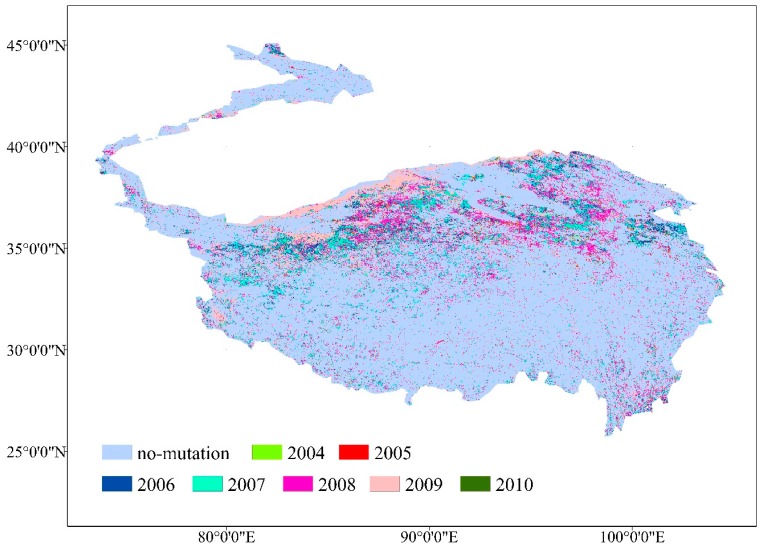
Change-points of VFC spatial distribution.

**Figure 10 ijerph-16-04709-f010:**
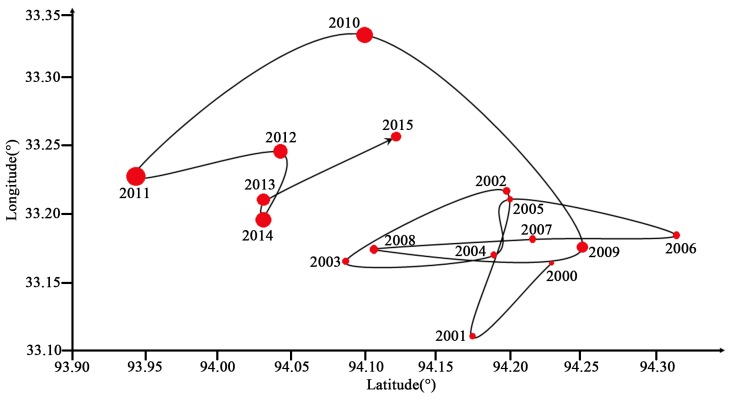
Center of gravity shift of the VFC during 2000–2015.

**Figure 11 ijerph-16-04709-f011:**
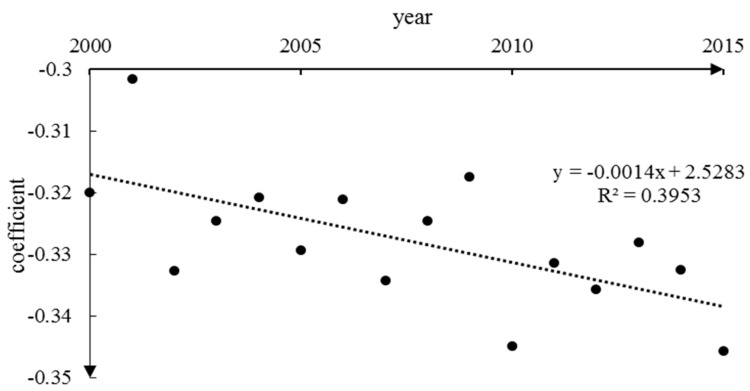
Correlation between elevation and the VFC.

**Figure 12 ijerph-16-04709-f012:**
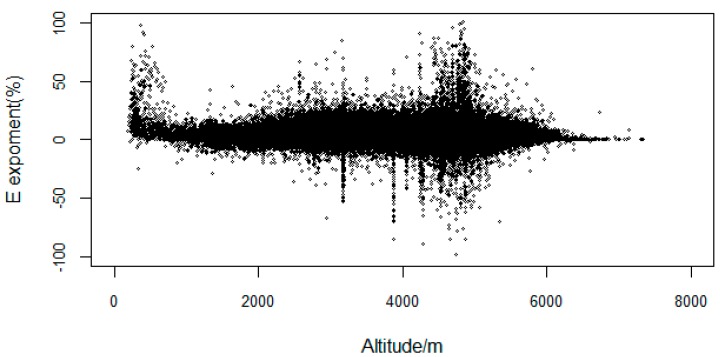
E exponent of the VFC at different elevations.

**Figure 13 ijerph-16-04709-f013:**
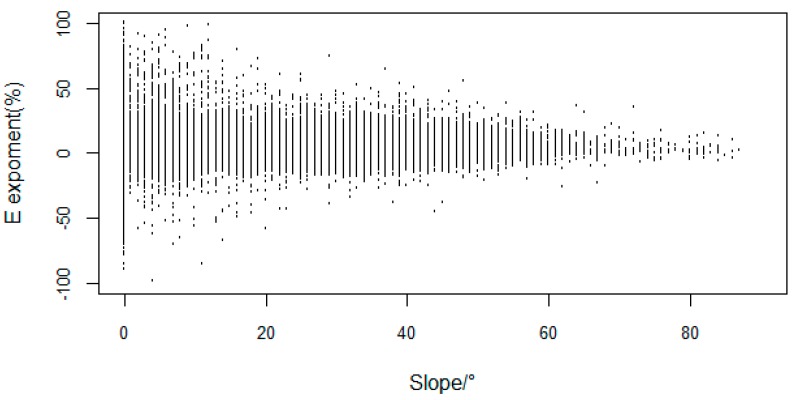
E exponent of the VFC for different slopes.

**Figure 14 ijerph-16-04709-f014:**
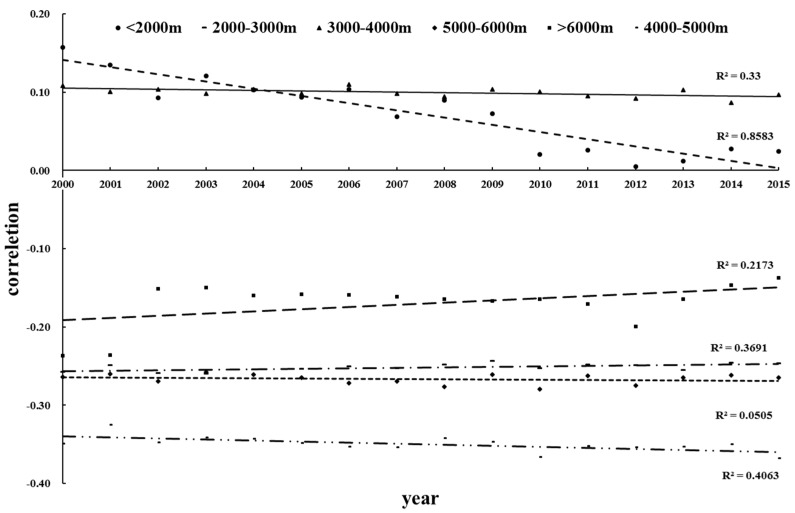
Correlation between the VFC and elevation under elevation grading.

**Figure 15 ijerph-16-04709-f015:**
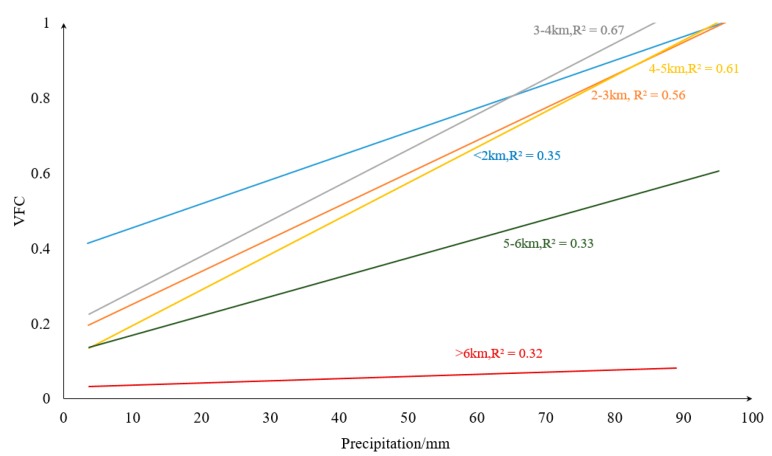
Relationship between precipitation and the VFC under different elevation classifications.

**Figure 16 ijerph-16-04709-f016:**
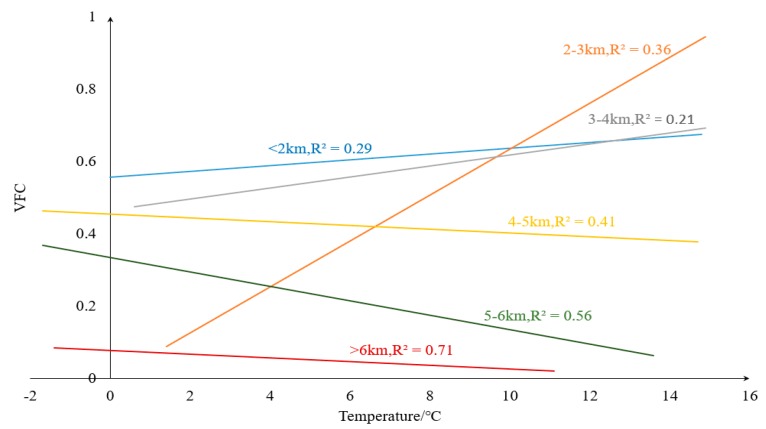
Relationship between temperature and the VFC under different elevation classifications.

**Table 1 ijerph-16-04709-t001:** Eco-geological division of the Tibetan Plateau.

Temperature Zone	Dry and Wet Area	Code	Eco-Geographical Division
HI	B	HIB1	High-cold shrub meadow area in the Guoluahang Plateau
	C	HIC1	Tibetan Plateau wide valley alpine meadow steppe
		HIC2	Quthai Lake Basin Alpine Grassland Area
	D	HID1	Kunlun high mountain plateau desert area
HII	AB	HIIAB1	Chuan Tibet eastern alpine valley coniferous forest
	C	HIIC1	Qingdong Qilian high mountain basin
		HIIC2	Highland, Shrubland Steppe
	D	HIID1	Desert Area in Qaidam Basin
		HIID2	Desert area in the north wing of the Kunlun Mountains
		HIID3	Ali mountain desert area
V	A	VA5	Yunnan Plateau
		VA6	East Himalayan South Wing
II	C	IID5	Yili Basin
		IID3	Junggar Basin
III	D	IIID1	Tarim and Turpan Basin

HI, Plateau sub-frigid zone; HII, Plateau temperate; V, Central Asia Subtropical; II, Middle Temperate; III, Warm temperate zone; A, humid area; B, semi-humid area; AB, humid semi-humid area; C, semi-arid area; D, Arid area

**Table 2 ijerph-16-04709-t002:** Regional statistics of the Tibetan Plateau.

Code	Area/km²	VFC			*p*
		Mean	2000	2015	
HID1	22.72	0.14	0.12	0.15	0.0007 (<0.001)
HIID1	29.77	0.17	0.15	0.19	0.0019 (<0.001)
HIID2	17.26	0.18	0.16	0.19	0.0018 (<0.001)
HIID3	7.34	0.21	0.20	0.23	0.0103 (<0.05)
HIC2	45.68	0.28	0.26	0.30	0.0016 (<0.01)
HIC1	17.9	0.38	0.36	0.38	0.0133 (<0.05)
IIID1	5.75	0.40	0.40	0.42	0.1373 (>0.05)
HIIC2	18.06	0.41	0.40	0.40	0.1151 (>0.05)
IID3	3.14	0.47	0.44	0.49	0.0274 (<0.05)
IID5	4.25	0.63	0.63	0.64	0.6204 (>0.05)
HIIC1	17.78	0.66	0.62	0.68	0.0004 (<0.001)
HIB1	27.30	0.69	0.68	0.69	0.2241 (>0.05)
HIIAB1	37.43	0.70	0.69	0.71	0.0217 (<0.05)
VA6	2.70	0.82	0.80	0.84	0.0001 (<0.001)
VA5	4.04	0.85	0.82	0.86	0.0003 (<0.001)

According to the eco-geographical division, the mean, initial, and final values of the TP area’s multi-year VFC are listed and sorted according to the mean.

**Table 3 ijerph-16-04709-t003:** Annual vegetation fractional coverage (VFC) variation over Tibet.

Significant Degree	The Degree of Vegetation Change	The Percentage of Total Area (%)	Area (km^2^)
*p* < 0.05	Slightly improvement	13.89	362,529
*p* < 0.01	Moderate improvement	10.71	2,610,000
*p* < 0.001	Severe improvement	4.18	109,098
*p* < 0.05	Stable	1.80	46,980
*p* < 0.05	Slightly degradation	0.07	1827
*p* < 0.01	Moderate degradation	0.30	7830
*p* < 0.001	Severe degradation	0.09	2349
*p* > 0.05	No significant	68.23	1,780,803

Classification of VFC change trends in the TP from 2000 to 2015, and area ratio statistics.

**Table 4 ijerph-16-04709-t004:** Corresponding statistics of climatic factors on the seasonal scale of VFC.

Climatic Factors	Correlation (*p* < 0.05)		Spring	Summer	Autumn	Winter
Precipitation	positive	Area ratio (%)	2.79	10.48	0.74	2.83
		Concentrated eco-geographical division	HIB1/HIIC1/HIC1	HIIC1/HIID1/HID1	HIIA/B1	HIID3/HIIC2
	negative	Area ratio (%)	4.72	1.98	4.17	2.12
		Concentrated eco-geographical division	HIID3/HIC2/HID1/HIB1/HIIA/B1	Less and discrete	HIC2/HIC1/HIB1	HIC2/HIB1
Temperature	positive	Area ratio (%)	27.42	10.32	39.21	0.31
		Concentrated eco-geographical division	HID1/HIC2/HIB1/HIIA/B1/HIIC1	HID1/HIC1/HIIC1/HIB1	HIID3/HIC2/HID1/HIC1/HIIC1	Less and discrete
	negative	Area ratio (%)	0.18	3.80	0.17	1.79
		Concentrated eco-geographical division	Less and discrete	HIC2/HIIC2	Less and discrete	HIID3
Sunshine duration	positive	Area ratio (%)	1.70	6.78	1.91	10.30
		Concentrated eco-geographical division	HIIC1/HIB1	HIB1/HIIA/B1/VA5/VA6	HIB1	HID1/HID1/HIID2
	negative	Area ratio (%)	0.93	4.36	0.72	2.21
		Concentrated eco-geographical division	HIB1/HIIA/B1	HIID1/HIIC1	IID3/IID5	HIIC2/HIIA/B1/VA5

Seasonal: spring (March–May), summer (June–August), autumn (September–November), and winter (December–February).

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
