# Peer review of "Vegetation Change and Its Relationship with Climate Factors and Elevation on the Tibetan Plateau"

_ijerph, 2019, doi:10.3390/ijerph16234709_

Round 1
Reviewer 1 Report
GENERAL REMARKS
Overall, I had a positive impression of this work; it is clear and rather well written (although it would greatly benefit from a general review of the English), with clear objectives and results.
My only concern is that the time range explored, 15 years, is rather short for a solid analysis of VCF time variation, but I understand that this is the range available with the MODIS dataset, and in any case the results are of a certain value for the interested audience.
The real problem, in my opinion, is that this article does not meet the journal’s scope. The focus of IJERPH is on environmental research in connection with public health issues, and in general with everything may be considered “environmental quality” under humans’ life point of view.
The article reports a study about vegetation variation at a large spatial scale, which may have an interest in the wider field of ecology and climate change, but not for this journal.
I therefore regret to say that the article is not acceptable for IJERPH.
Anyway, I hearthily encourage the authors to re-submit this work to another journal with a more adequate scope. There are many possible options, since this work could fit with journals dealing with remote sensing of environment, ecology or climate change.
Here below I add some observations which may be of help for a resubmission.
Lines 86-87
It should be specified that the total solar radiation value refer to a year.
Lines 169-174
The terminology presented here is rather odd and prone to confusion.
I suggest to use the word “pixel” to indicate a single point of the image, and “grid” to indicate the two-dimension array formed by the pixels (which corresponds to the image itself).
Line 172
“Pettitt” instead of “Pettit”
Lines 173-174
Each R package should be correctly cited and citations must be reported in the reference list
Line 187 (Table 2)
I think it is preferable to report the actual P value rather than simply its position respect to threshold values (i.e. write for example “0.087” and not simply “>0.05”). If you want to mark the significant values, you can underline them or use bold characters.
The Table could be more readable if you rank results in increasing (or decreasing) order respect to VFC variation.
Line 229 (Figure 6)
The legend of figures is hardly readable. It seems to me that the legend is the same for all figure, so you may keep just one legend, in a larger size, and cancel all the others.
Line 292 (Figure 7)
It is not quite clear how Figure 7 was built, so the message contained in it is difficult to grasp. The description (lines 288-292) should be made clearer.
Line 371 (Figure 9)
In the figure legend, the years are written with a comma, which it shouldn’t be.
Author Response
Dear editor and reviewers:
Thank you for all your useful comments and kind suggestions on our manuscript. We have revised the manuscript accordingly, and all the revision are marked in red. Detailed revisions are listed below: Answers to reviewers:
Reviewer#1
Lines 86-87 It should be specified that the total solar radiation value refer to a year.Answers: Thanks for your suggestion and we have revised in line 93.
Lines 169-174 The terminology presented here is rather odd and prone to confusion. I suggest to use the word “pixel” to indicate a single point of the image, and “grid” to indicate the two-dimension array formed by the pixels (which corresponds to the image itself).Answers: Thanks for your suggestion and we have revised in line 175-176.
Line 172“Pettitt” instead of “Pettit”Answers: Thanks for your suggestion and we have revised the full manuscript.
Lines 173-174 Each R package should be correctly cited and citations must be reported in the reference list.Answers: Thanks for your suggestion, am not sure about the format of the reference program, but it has been revised and added to the end of reference.
I think it is preferable to report the actual P value rather than simply its position respect to threshold values (i.e. write for example “0.087” and not simply “>0.05”). If you want to mark the significant values, you can underline them or use bold characters. The table could be more readable if you rank results in increasing (or decreasing) order respect to VFC variation.Answers: Thanks for your suggestion and we have revised the Table 2 in line 191.
Line 229 (Figure 6) The legend of figures is hardly readable. It seems to me that the legend is the same for all figure, so you may keep just one legend, in a larger size, and cancel all the others.Answers: Thanks for your suggestion and we have revised in line 243.
Line 292 (Figure 7) It is not quite clear how Figure 7 was built, so the message contained in it is difficult to grasp. The description (lines 288-292) should be made clearer.Answers: Thanks for your suggestion, the modifications and additions have been made to line 255-259.
Line 371 (Figure 9) In the figure legend, the years are written with a comma, which it shouldn’t be.Answers: Thanks for your suggestion and we have revised the Figure 9 in line 345.

Reviewer 2 Report
Dear authors,
This manuscript titled “Vegetation Change and Its Relationship with Climate Factors on the Tibetan Plateau”. The authors try to analyze the vegetation change characteristics and the correlation of vegetation change with climate factors through the Normalized Difference Vegetation Index datasets of the TP for 2000 –2015, based on a statistical and Mann–Kendall trend analyses. It’s an interesting work but needs intensive revision before it goes into the final shape of an article format. I have the following comments and hope these comments will be useful to improve your manuscript.
Abstract section
Line 25 the sentence is not clear – this finding of this study is important for which region? (for only this study area or the world). Can we also replicate to some other areas?
Line 27: key-worlds should be different from the title of the manuscript (Tibetan Plateau should be removed from key-worlds)
Introduction and Methodology sections
Line 100: “2.1.1. Study Area Division” had put the wrong place? Please check
Line 102-104: Please explain how the authors can divide the study area into 15 sub-regions (which data, which year were used)
Line 104 check citation of MDPI style
Line 122: Use the colon very arbitrarily so need to revise it
Line 135 check citation of MDPI style
Line 130: What types of climatic data were used (hourly, daily, yearly)?
Line 140-141: NDVI max and VDVI min need to redefine
Line 127: check data period 2000-2016 while all results show from 2000-2015 which one is correct please make it clearly.
Results section
Figures 4 and 6: Legends I would not read it please improve it so please go through the text to improve all Figures.
Line 297 and 304 “Figure” and go through the text to correct the similar issues
Line 149: I would not see Z in any formulas
Line 177-179: “generally decreased from southeast to northwest: 60.3% of the VFC in the whole area increased and 178 38.7% decreased. The significantly (P < 0.05) increased area accounted for 28.7%” what did you mean? Make it clear, please.
Line 235: the authors said that “Correlation between the VFC and precipitation in most parts of the TP were not significant (P > 0.05; Fig. 6a). Correlation coefficients had clear differences in seasonal and spatial variations”. Explain why don’t you try to the relationship at seasonal scale and sub-regions instead for the whole year and whole study area? If not I think do you need to say more about that “correlations between the VFC and precipitation were highest in summer,….” If so you can explain why different among season?
Why did you consider precipitation, temperature, and sunshine duration as climatic data? Were you based on previous studies? If so please show it in the Introduction section.
Subsection 3.3 (winter, autumn, summer, spring) and subsection 3.4 (dry and wet season). Are there many types of the season in your study area? I was confused.
Line 288-289: why you need to consider precipitation again while it has no significant with VFC on subsection 3.3. Sometimes you need to explain why did you need to do like this or like that because of ….
Line 376: subsection 4.2: “Vegetation variation with topography” while this title of manuscript focused on climatic factors?
Discussion and conclusion:
I would not see you discuss and conclusion about the relationship between climatic factors and VFC. Finally, did you found which climatic factor/Topographical elements?
Author Response
Dear editor and reviewers:
Thank you for all your useful comments and kind suggestions on our manuscript. We have revised the manuscript accordingly, and all the revision are marked in red. Detailed revisions are listed below: Answers to reviewers:
Reviewer#2
Line 25 the sentence is not clear – this finding of this study is important for which region? (for only this study area or the world). Can we also replicate to some other areas?Answers: Thanks for your suggestion and we have revised in line 27-28.
Line 27: key-words should be different from the title of the manuscript (Tibetan Plateau should be removed from key-worlds)Answers: Thanks for your suggestion and we have removed from the key-words.
Line 100: “2.1.1. Study Area Division” had put the wrong place? Please checkAnswers: Thanks for your suggestion, we have modified and checked all the titles.
Line 102-104: Please explain how the authors can divide the study area into 15 sub-regions (which data, which year were used)Answers: Thanks for your suggestion, the modifications and additions have been made to line 109-111.
Line 104 check citation of MDPI styleAnswers: Thanks for your suggestion and we have revised in line 112.
Line 122: Use the colon very arbitrarily so need to revise itAnswers: Thanks for your suggestion and we have revised in line 128-129.
Line 135 check citation of MDPI styleAnswers: Thanks for your suggestion and we have revised in line 140.
Line 130: What types of climatic data were used (hourly, daily, yearly)?Answers: Thanks for your suggestion, the modifications and additions have been made in line 137.
Line 140-141: NDVI max and VDVI min need to redefineAnswers: Thanks for your suggestion and we have revised in line 144-145.
Line 127: check data period 2000-2016 while all results show from 2000-2015 which one is correct please make it clearly.Answers: Thanks for your suggestion and we have revised in line 132.
Figures 4 and 6: Legends I would not read it please improve it so please go through the text to improve all Figures.Answers: Thanks for your suggestion and we have revised in line 213 (Figures 4) and line 242 (Figures 6).
Line 297 and 304 “Figure” and go through the text to correct the similar issuesAnswers: Thanks for your suggestion and we have revised the full manuscript.
Line 149: I would not see Z in any formulasAnswer: Thank you for your suggestion, we have added the formulas in line 152-154.
Line 177-179: “generally decreased from southeast to northwest: 60.3% of the VFC in the whole area increased and 178 38.7% decreased. The significantly (P < 0.05) increased area accounted for 28.7%” what did you mean? Make it clear, please.Answers: Thanks for your suggestion, and modifications and additions have been made to line 183 -185.
Line 235: the authors said that “Correlation between the VFC and precipitation in most parts of the TP were not significant (P > 0.05; Fig. 6a). Correlation coefficients had clear differences in seasonal and spatial variations”. Explain why don’t you try to the relationship at seasonal scale and sub-regions instead for the whole year and whole study area? If not I think do you need to say more about that “correlations between the VFC and precipitation were highest in summer,….” If so you can explain why different among season?Answers: Thanks for your suggestion, the annual scale of vegetation and climate over the TP has been supplemented in the Introduction section. The effects of precipitation to vegetation on seasonal scale has supplemented in 4.2.
Why did you consider precipitation, temperature, and sunshine duration as climatic data? Were you based on previous studies? If so please show it in the Introduction section.Answers: Thanks for your suggestion and we have revised in Introduction section.
Subsection 3.3 (winter, autumn, summer, spring) and subsection 3.4 (dry and wet season). Are there many types of the season in your study area? I was confused.Answers: Thanks for your suggestion, we added the time-scale division in 2.1.2.
Line 288-289: why you need to consider precipitation again while it has no significant with VFC on subsection 3.3. Sometimes you need to explain why did you need to do like this or like that because of ….Answers: Thanks for your suggestion, and this part is supplemented in the introduction and 3.4.
Line 376: subsection 4.2: “Vegetation variation with topography” while this title of manuscript focused on climatic factors?Answers: Thanks for the suggestion, the title and content of 4.2 have been revised.
I would not see you discuss and conclusion about the relationship between climatic factors and VFC. Finally, did you found which climatic factor/Topographical elements?Answers: Thanks for the suggestion, a discussion of vegetation and climate was added in 4.2. The main topographical element is elevation, and we revised the discuss in the 4.2.

Reviewer 3 Report
Major comment:
The study was able to trace the long-term vegetation changes in Tibetan Plateau and how human interventions and ecological engineering efforts helped shape the current vegetation cover of the area. It also detects the relationship of climate to these changes and contrast the wet and the dry periods. That is very impressive. I can understand the results. However, in order to improve the manuscript, I feel that the discussion section fell a little weak for me. Its still explaining more of the results but not more on the implications, the cause and effect of climate, human intervention and the like. Maybe cut short some of the very detailed explanation of the result that readers can already see in the tables and figures but exert more effort how does your methods/results can be used as a tool to picture out the current situation of the Tibetan Plateau. Impress the readers with what you got without necessarily drowning the readers with your technical analysis. Be straightforward on what you want to convey out of this study. The storyline must be clear. For me, the the manuscript fails to discuss in-depth analysis of the results to provide the reader the take home message this study would like to present. Maybe work more on this aspect. Overall, the results were great!
Minor suggestions:
Line 87: at least give a temperature range, if possible.
Line 144: The change range of vegetation coverage. What does this mean? Is this the rate of change?
Line 187: This table and for all tables in the manuscript should be considered a ‘stand alone’. It means readers can understand the table without going through the pages in the manuscript to understand clearly what the table is all about. Please provide a clear description in all tables.
Line 196: All figures in this manuscript are all impressive. Just make sure to have a very clear figure text, legend and figure caption. The captions, just like tables shall also be stand-alone, hence, careful description but brief and concise is needed. In this figure 3, the x and y axes labels seemed pixelated.
Line 209: Figure 4 legends were not clear enough. The quality has to be improved.
Line 230: The figure legend and axes labels were hardly recognizable.
Line 234: I suggest the seasonal correlations between VFC and climatic factors from section 3.3.1 to 3.3.3 can be presented in a tabular form arranged categorically by geographic location or any arrangement easily understandable. These sections occupied a lot of space just presenting the results to where you could have presented it in one table. The space saved from these sections can be used instead to exhaustively discuss the implications about the results rather than just a mere presentation of results.
Line 294: Figure 7. Maybe use () in the units rather (/) e.g. Precipitation (mm) rather than Precipitation/mm and so and so forth.
Lines 306-309: Its not clear, at least for me, how this sunshine duration was obtained? When you say in the x-axis 100 – 300 h sunshine duration, what does it tells me? sunshine from what duration? Is it the time of the day? Accumulated sunshine hour? So if you have your dataset from 2000 – 2015, where does this 100 – 300h fall on this duration? Sorry for failure to understand this part.
Line 326: Figure 8 (purple in the legend) connotes double factors but as to what climate or what combination among Precipitation, temperature and sunshine duration?
Line 328: What is mutation situation? Maybe explain a little what is this about? How one event can be situated as mutation?
Line 393: ‘significantly’?
Author Response
Dear editor and reviewers:
Thank you for all your useful comments and kind suggestions on our manuscript. We have revised the manuscript accordingly, and all the revision are marked in red. Detailed revisions are listed below: Answers to reviewers:
Reviewer#3
Line 87: at least give a temperature range, if possible.Answers: Thanks for your suggestion and we have revised in line 94.
Line 144: The change range of vegetation coverage. What does this mean? Is this the rate of change?Answers: Thanks for your suggestion and we have revised in line 148, and the E refers to the vegetation change rate of 16 years from 2000 to 2015, and the range is [-100%, +100%].
Line 187: This table and for all tables in the manuscript should be considered a ‘stand alone’. It means readers can understand the table without going through the pages in the manuscript to understand clearly what the table is all about. Please provide a clear description in all tables.Answers: Thanks for your suggestion and all the tables have been remarked in the below to guide the reader to better understand the information.
Line 196: All figures in this manuscript are all impressive. Just make sure to have a very clear figure text, legend and figure caption. The captions, just like tables shall also be stand-alone, hence, careful description but brief and concise is needed. In this figure 3, the x and y axes labels seemed pixelated.Answers: Thanks for your suggestion, we have revised Figure 3 in line 201 and all the legends and texts of figures have been adjusted to ensure that readers can read clearly.
Line 209: Figure 4 legends were not clear enough. The quality has to be improved.Answers: Thanks for your suggestion and we have revised in line 213.
Line 230: The figure legend and axes labels were hardly recognizable.Answers: Thanks for your suggestion and we have revised Figure 6 in line 242.
Line 234: I suggest the seasonal correlations between VFC and climatic factors from section 3.3.1 to 3.3.3 can be presented in a tabular form arranged categorically by geographic location or any arrangement easily understandable. These sections occupied a lot of space just presenting the results to where you could have presented it in one table. The space saved from these sections can be used instead to exhaustively discuss the implications about the results rather than just a mere presentation of results.Answers: Thanks for your suggestion and we have revised in line 247, and the discussion about climatic factors with vegetation on seasonal scale has supplemented in 4.2.
Line 294: Figure 7. Maybe use () in the units rather (/) e.g. Precipitation (mm) rather than Precipitation/mm and so and so forth.Answers: Thanks for your suggestion and we have revised in line 260.
Lines 306-309: Its not clear, at least for me, how this sunshine duration was obtained? When you say in the x-axis 100 – 300 h sunshine duration, what does it tells me? sunshine from what duration? Is it the time of the day? Accumulated sunshine hour? So if you have your dataset from 2000 – 2015, where does this 100 – 300h fall on this duration? Sorry for failure to understand this part.Answers: Thanks for your suggestion and we have revised. The original data of the 2000-2015 sunshine duration used in the study is the monthly data, which is the sum of the sunshine hours of a month, and belongs to a part of the meteorological observation data (including the monthly maximum temperature, the monthly minimum temperature, the monthly sunshine duration, and the monthly average Precipitation, etc), and downloaded on China Meteorological data sharing service network.
Line 326: Figure 8 (purple in the legend) connotes double factors but as to what climate or what combination among Precipitation, temperature and sunshine duration?Answers: Thanks for your suggestion, the word ‘combination’ may be ambiguous here. The main purpose is to find areas that are sensitive to climate factors during the growing season. Among the precipitation, temperature, and sunshine duration, the area where vegetation responds in one case considered to be insensitive. The double factors area is a region where any two climatic factors can change the vegetation and is relatively sensitive. The triple factor zone is the zone where all three climatic factors can affect vegetation changes and is the most sensitive zone, and we have revised in line 283-287.
Line 328: What is mutation situation? Maybe explain a little what is this about? How one event can be situated as mutation?Answers: Thanks for your suggestion, and the words ‘mutation’ may be ambiguous here and we have modified to ‘change points’ in the title in line 298. The ‘Pettitt change-point test’ is to identify and extract the corresponding maximum value in a time series and the calculation method introduced in lines 169-170.

Round 2
Reviewer 1 Report
Dear Authors,
after reviewing the revised version of the article, I confirm the overall positive impression without any further modification request.
I definitely recommend the article for publication.
Author Response
Thank you for your suggestions and comments on our manuscript. It is very meticulous. In the process of future research, we will still according to your suggestions.

Reviewer 2 Report
The manuscript includes the text and Fig have significantly improved but I have minor revision please see below,
Line 110, 134, 140: check the MDPI website citation and go through all the text to revise them.
Line 653: check citation
line 429-430 and Fig 14 show the relationship between VFC and elevation so your hypothesis elevation is one of the climate factors (also see the title of manuscript)?
Author Response
Dear editor and reviewers:
Thank you for all your useful comments and kind suggestions on our manuscript. We have revised the manuscript accordingly, and already asked the editor for help on some of the issues in the reference format and made modification. and all the revision are marked in red. Detailed revisions are listed below:
Answers to reviewers:
Reviewer#2
Line 110, 134, 140: check the MDPI website citation and go through all the text to revise them.Answers: Thanks for your suggestion and we have revised,and if the website reference format still has an error, I will contact the editor to continue the modification.
Line 653: check citationAnswers: Thanks for your suggestion and we have revised in line 664.
line 429-430 and Fig 14 show the relationship between VFC and elevation so your hypothesis elevation is one of the climate factors (also see the title of manuscript)?Answers: Thanks for your suggestion, and we have modified the title of the manuscript and revised in lines 432-435. The temperature changes with the elevation, which in turn affects the growth of vegetation to some extent. As a geographical factor, the elevation indirectly affects the coverage of vegetation, and the vegetation shows a certain regularity with the elevation change.
